# Bias and Class Imbalance in Oncologic Data—Towards Inclusive and Transferrable AI in Large Scale Oncology Data Sets

**DOI:** 10.3390/cancers14122897

**Published:** 2022-06-12

**Authors:** Erdal Tasci, Ying Zhuge, Kevin Camphausen, Andra V. Krauze

**Affiliations:** 1Center for Cancer Research, National Cancer Institute, NIH, Building 10, Bethesda, MD 20892, USA; erdal.tasci@nih.gov (E.T.); zhugey@mail.nih.gov (Y.Z.); camphauk@mail.nih.gov (K.C.); 2Department of Computer Engineering, Ege University, Izmir 35100, Turkey

**Keywords:** class imbalance, oncology, machine learning, artificial intelligence, clinical data

## Abstract

**Simple Summary:**

Large-scale medical data carries significant areas of underrepresentation and bias at all levels: clinical, biological, and management. Resulting data sets and outcome measures reflect these shortcomings in clinical, imaging, and omics data with class imbalance emerging as the single most significant issue inhibiting meaningful and reproducible conclusions while impacting the transfer of findings between the lab and clinic and limiting improvement in patient outcomes. When employing artificial intelligence methods, class imbalance can produce classifiers whose predicted class probabilities are geared toward the majority class ignoring the significance of minority classes, in turn generating algorithmic bias. The inability to mitigate this can guide an AI system in favor of or against various cohorts or variables. We review sources of bias and class imbalance and relate this to AI methods. We discuss avenues to mitigate these and propose a set of guidelines aimed at limiting and addressing data and algorithmic bias.

**Abstract:**

Recent technological developments have led to an increase in the size and types of data in the medical field derived from multiple platforms such as proteomic, genomic, imaging, and clinical data. Many machine learning models have been developed to support precision/personalized medicine initiatives such as computer-aided detection, diagnosis, prognosis, and treatment planning by using large-scale medical data. Bias and class imbalance represent two of the most pressing challenges for machine learning-based problems, particularly in medical (e.g., oncologic) data sets, due to the limitations in patient numbers, cost, privacy, and security of data sharing, and the complexity of generated data. Depending on the data set and the research question, the methods applied to address class imbalance problems can provide more effective, successful, and meaningful results. This review discusses the essential strategies for addressing and mitigating the class imbalance problems for different medical data types in the oncologic domain.

## 1. Introduction

Large-scale medical data carries significant areas of underrepresentation and bias at all levels: clinical (demographic, disease characteristics), biological (heterogeneity of disease), management and outcomes [1,2,3,4]. Underrepresentation or overrepresentation of one or several categories as related to clinical and disease characteristics, management or outcomes, and resulting imaging studies and biospecimens results in data set bias and class imbalance, or unequal distribution of features in a provided data set. This is often exemplified in oncologic data sets by, for example, overrepresentation of patients with superior performance status on clinical trials or, for example, underrepresentation of certain demographics such as the inclusion of certain racial and ethnic groups or patient populations who experience greater challenges with access to care. These uneven distributions within data sets result in a majority class, or the subset of the data that is overrepresented, and a minority class, or a subset of the data that is underrepresented. Class imbalance in clinical, imaging, and omics data sets is the rule, not the exception [5]. In the context of clinical data, this may be the result of the type of practice setting (e.g., publicly funded systems vs. academic vs. private practice), location (urban vs. rural, affluent vs. limited resources), data acquisition (manual vs. batch collection from electronic medical records), data curation (handling missing and inaccurate data). This heterogeneity in “big data” is reflected in publicly available data sets in all types of medical settings and clinical contexts [6,7]. The number and importance of publications about the class imbalance problem have increased over time (see Figure 1) [8]. The variability of data at all levels and its presence in large-scale data sets of all subtypes is emerging as the single most significant issue inhibiting the harnessing of data for meaningful and reproducible conclusions. It impacts the transfer of findings between the lab and the clinic and limits improvement in patient outcomes [9]. However, obtaining balanced data in medical data sets of all subtypes is challenging secondary to limitations in patient numbers, cost, privacy and security of data sharing, and the complexity of generated data, resulting in significant efforts to achieve representative data sets [10,11]. For example, in medical image data sets, the number of one class samples generally exceeds the number of other class samples, causing class imbalance problems which are increasingly a hot topic in machine learning and data mining [10]. While the class with fewer examples represents the minority class, another class represents the majority class. Machine learning models/ artificial intelligence (AI) systems considering all data, therefore, produce biased classifiers whose predicted class probabilities are geared toward the majority class ignoring the significance of minority classes [12,13]. The algorithmic bias here is defined as the prejudice embedded in the decision resulting from an AI system in favor or against a person, group, or thing that is taken into account to be unfair [14]. In other words, algorithmic bias is the difference between the predicted value and the actual value of our model. If the bias is high, the model cannot capture the pattern (e.g., essential features) of data, it misses the relations between the features and targets, and the learning model cannot generate appropriate predictions on testing data. This situation is called underfitting. The decision boundaries created by traditional machine learning models tend to be biased toward the majority class; therefore, the minority classes are most likely misclassified for class imbalanced data sets.

Furthermore, the machine learning models treat minority classes as noise; therefore, noise may be incorrectly classified [15]. The degree of imbalance is defined as the ratio of the sample size of the minority class to that of the majority class [11]. The degree of imbalance in medical image data sets generally can be as low as 1:5, 1:10, 1:100, or 1:1000. We will review sources of bias in clinical data sets and their current impact on large scale data use, the effect of data bias and class imbalance on AI methods, and avenues to mitigate these, generating a set of guidelines aimed at limiting and addressing bias and class imbalance in large-scale medical data. The organization of this paper is as follows: Section 1: we discuss sources of data-based class imbalance problems in machine learning as relevant to medical applications (i.e., oncology). Additionally, we provide explanations for the origins of bias and class imbalance in large-scale medical data; Section 2: we discuss methods employed in addressing this problem; Section 3: we present guidelines to mitigate bias and address class imbalance in oncologic data sets toward transferrable AI; Section 4 concludes this paper with future directions.

## 2. Sources of Bias and Class Imbalance in Large Scale Medical Data

### 2.1. Gender and Race Imbalance

There is increasing awareness of bias and class imbalance sources in medical data sets. Notably, gender [16,17] and race bias [18,19] with class imbalances are increasingly highlighted in the literature. Both factors have been shown to affect outcomes. They have also been shown to affect data collected at all levels, clinical [6], imaging [13], and biospecimen [20], and in turn, analysis of downstream genomic and proteomic data sets (Figure 2) [8]. Oncologic disciplines where this has been highlighted include large patient volume oncologic sites such as lung cancer [20], breast cancer [18], and head and neck cancer [21] but also lower volume sites such as central nervous system malignancies [22,23].

Nonetheless, in complex data sharing performed with limited resources, a significant proportion of oncologic literature employs large-scale data or uses publicly available data sets for validation, using a biased data set. The use of “easy data” is highly prevalent and results in AI often not transferrable or scalable. Many AI approaches do not employ clinical data or only minimal clinical data, preferring to rely on only imaging or only omic data analyzed with semi-supervised or unsupervised methods. Validation with larger data sets with fewer missing data is carried out in an effort to limit bias. This approach can result in conclusions that have unclear or limited clinical applicability and, due to class imbalance and overfitting, are not replicated in other data sets. Classification of data can improve the outcome of AI approaches but is impacted by the employed data with inter and intra-observer variability.

### 2.2. Limited Capture and Inclusion of Social Determinants of Health (SDOH) and Disability

Additional significant sources of data-related bias involve class imbalance due to the lack of capture and inclusion of social determinants of health (SDOH) [24] and patient disability in AI methods. Social determinants of health consist of the geographical and socioeconomic conditions where people are born, live, and age, and the drivers of these conditions with cancer outcome disparities reflecting these aspects [24]. Disability affects the aggressiveness and accessibility of oncologic management but can be challenging to collect consistently in data sets and is intersectional with other factors noted above, such as gender and SDOH. Both SDOH and disability have been particularly challenging to capture robustly in data sets and, as a result, are often not included in AI analysis [24,25]. Class imbalance carries significant implications concerning these domains, and studies have looked at computational approaches, including natural language processing in unstructured data to expand data sets to capture disability as a feature [26].

### 2.3. Variability in Oncologic Management

Patient management in oncology can vary based on several clinical and disease-related factors, including features discussed in Section 1 and Section 2 above and the evolution of staging and treatment paradigms in any one condition over time. Notably, demographics and management can be drastically different between trial populations and the real world [27,28], resulting in heterogeneous data mirrored in data sets employed for analysis, training, testing, and validation (Figure 2). In real-world settings, oncologic management can be highly variable based on geographical location, access, and resources, which can significantly impact training AI algorithms and result in non-transferable products.

### 2.4. Variability in Oncologic Outcomes Capture

Outcome measures of most studies depend on capturing progression and survival parameters with patient- and physician-reported outcomes increasingly being implemented. When patients are treated in clinical trials, these parameters are more rigorously captured than patient outcome capture in real-world settings. In real-world settings, patient survival (date of diagnosis to the date of death) is a robust parameter (albeit with the limitation of the cause of death being inconsistently captured). However, the date of disease progression is inconsistently available, suffering from difficulties in determining progression (clinical, radiographic, biochemical) and resource limitations to data annotation. Protocols are also attempting to address this aspect with adaptive trial designs that are increasingly complex. Provided the growing cost of developing and testing oncologic management avenues, the ability to identify subpopulations of patients in whom treatment is effective vs. those where early intervention or alteration in management may be beneficial is driven by robust data that originates from highly curated trial populations. Computerized patient-reported outcome measures and patient-generated data are on the rise [29] in clinical settings, although such large-scale data are likely to intersect with other sources of data imbalance. As a result, large-scale medical data are less likely to originate from study patients (which is a source of class imbalance in and of itself) and more likely to grow using real-world data (also a potential source of class imbalances).

### 2.5. Knowledge Translation Bias

While many a conversation involves direct data bias, as described above, it should be noted that data sets, their analysis, and prediction results, as can be accessed in the public domain, lead back to published results. However, most published studies report positive findings and far less report on unsuccessful AI as exemplified in the context of, for example, skin cancer classification [30,31], sepsis prediction [32], the use of race correction in clinical algorithms [33], and notably the COVID-19 pandemic [34]. Publication bias is well described reflecting both the excitement and the fundamental clinical gaps in aspects of diagnosis, response, or management that need to be overcome with the perceived advantage of artificial intelligence approaches. Negative studies have the potential to be highly instructive but lack publication exposure. The reason for unsatisfactory results may relate to a lack of capture and adaptation to class imbalance sources, limiting the exposure of such studies to other investigators, and the development of meaningful solutions to flawed data.

## 3. Overview of the Existing Methods for the Class Imbalance Problem

Many studies have been conducted in the literature to address the problem of bias and class imbalance and increase the learning performance of the model utilized. Although there is no universal method to deal with the class imbalance problem, researchers have suggested various methods and approaches for this, according to the size of the data set they use, data distribution, imbalance ratio, and performance criteria of the model. According to the methods used, the existing approaches can be mainly divided into a data level, algorithm level, and hybrid. Additionally, these levels can be grouped into various techniques (i.e., re-sampling, recognition-based approach, cost-sensitive learning, and ensemble learning) [11]. An overview of the existing methods for the class imbalance problem is illustrated in Table 1 and Figure 3. This review summarizes the current technical tools for handling class imbalance problems, particularly medical image data sets. We mainly focus on image-centric issues and the strategies for addressing these in this study. The advantages and drawbacks of the methods employed in the literature are described in the following sections.

### 3.1. Data-Level Methods

Data level methods can be re-sampling methods such as over-sampling and under-sampling (Figure 3). They aim to balance data sets by utilizing re-sampling methods such as increasing the sample size of the minority classes (over-sampling) or reducing the sample size of the majority class (under-sampling) [15,65]. In some cases, over-sampling methods may cause overfitting due to duplicating a minority of samples for the related data set; under-sampling also may have some disadvantages, such as discarding useful information for the data distribution [15]. Under-sampling can remove valuable and significant patterns; therefore, it can cause the loss of useful information [10]. Both over-sampling and under-sampling methods can be applied separately or as a hybrid (e.g., SMOTE-ENN or SMOTE-Tomek [15]) to any learning model for class imbalance problems to handle these limitations and provide good generalization ability of the model. Both over-sampling and under-sampling methods can be applied separately or as a hybrid to any learning model for class imbalance problems. Data-level methods are a type of data preprocessing stage that redistributes training instances to establish well-balanced data sets. Data-level methods’ advantages are independent of underlying the learning model/classifier and can be easily performed [10]. However, under-sampling can remove valuable and significant patterns; therefore, it can cause the loss of useful information [10].

On the other hand, over-sampling results in a time-consuming process due to extra computational cost to the learning model used. To automate the incident triage and severity prioritization process for saving time and cost as well as overcoming errors in the radiation oncology incident learning system, Bose et al. applied convolutional neural networks (CNN) and long short-term memory (LSTM) with minority and random over-sampling strategies to handle high-class imbalanced data [35]. They employed incident reports obtained from Virginia Commonwealth University (VCU) and the US Veterans Health Affairs (VHA) radiation oncology centers. Brown et al. developed various data-filtering techniques, including under-sampling methodologies for imbalanced data classification methods, to provide automated radiation therapy (RT) quality assurance (QA) for prostate cancer treatment. Their proposed approach obtains strong performance results with high classification accuracy [36]. Liu et al. used Synthetic Minority Oversampling Technique (SMOTE) to overcome with class imbalance problem for Glioblastoma Multiforme (GBM) brain tumor images obtained from Moffitt Cancer Research Center and TCGA (The Cancer Genome Atlas Program) T1-Contrast data set (NIH’s Cancer Imaging Archive Dataset) [37]. Their method achieved 37.83% AUC (Area under the Curve) improvement for brain magnetic resonance imaging (MRI) images. In another study, Suarez-Garcia et al. also developed a basic model for glioma grading based on texture analysis using the under-sampling method to handle class imbalance data [38].

Furthermore, to predict the occurrence of liver cancer with skewed epidemiological data and tackle the class imbalance problem, Li et al. introduced two different under-sampling methods based on K-Means++ and learning vector quantization (LVQ) [39]. Isensee et al. developed a fast and effective deep learning-based segmentation method, namely nnU-net, which automatically configures all segmentation architecture without user interventions for arbitrary new tasks/data sets [40]. Their method employs an oversampling strategy by randomly selecting 66.7% of samples from the training data for a robust handling class imbalance problem. In case the re-sampling methods may not provide performance improvement of the prediction models, other methods such as ensemble learning (e.g., boosting) or various feature selection approaches can be employed.

### 3.2. Algorithm-Level Methods

Researchers can alter traditional machine learning models at the algorithm level, assigning weight or cost to a classifier to reduce bias towards the majority class. The learning model is not affected by the class distribution in this process. These methods may be recognition-based, cost-sensitive, or ensemble learning-based (Table 1 and Figure 3).

#### 3.2.1. Recognition-Based Methods

One-class learning is an approach that is used in the absence of non-target class instances for recognition-based methods. Outlier detection and novelty detection are examples of the one-class classification. This method provides high performance, particularly on high-dimensional data. Various models, such as support vector machine (SVM) and isolation forest, can be built by one class learning, while some models, such as decision trees and Naïve Bayes, cannot [41]. Gao et al. proposed a deep-learning-based one-class classification approach to handle imbalanced class data for medical images such as breast MRIs and space-occupying kidney lesion (SOKL) data sets [42]. For head and neck cancer radiation therapy outcome predictions, Welch et al. (2020) employed the isolation forest algorithm, an ensemble of isolation trees constructed to deal with the class imbalance and detect data anomalies [43].

#### 3.2.2. Cost-Sensitive Methods

Cost-sensitive methods are particularly critical for medical applications due to the importance of false positive and false negative instances. In this approach, the misclassification cost is adjusted to balance the majority and minority classes (e.g., a false negative prediction may be assigned a higher weight (i.e., cost) as compared to a false positive prediction [44]. This practical solution yields cost-sensitive learning for this problem. There are various cost-based approaches such as weighted cross-entropy, multiclass dice loss function, and focal loss to overcome the class imbalance in data in the literature.

The weighted cross-entropy loss equation [45] is defined as Equation (1). Cross-entropy loss function, called logarithmic loss, or log loss, is based on the Information Theory and related to the level of uncertainty (i.e., entropy) for the outcome of variables. Cross-entropy is used for adjusting model weights during the training stage to minimize this loss value, and this score is calculated by penalizing the probability based on how far the predicted value is from the true expected value. The smaller the loss indicates, the better the model.
(1)$$L_{w}=-\sum_{i=1}^{m}(\alpha y_{i}\log({\;\overset{⌢}{y}}_{i})+(1-y_{i})\log(1-{\;\overset{⌢}{y}}_{i}))$$
where α represents the imbalance parameter.

Another class loss function to handle class imbalance in data for the medical image segmentation is the multiclass dice loss function [46]. It is differentiable and can be easily adapted into various deep learning frameworks. The multiclass dice loss function is illustrated in Equation (2).
(2)$$L_{dc}=-\frac{2}{\left|K\right|}\sum_{k\in K}\frac{\sum {}_{i}u_{i,k}v_{i,k}}{\sum {}_{i}u_{i,k}+\sum {}_{i}v_{i,k}}$$
where *u* denotes the softmax output of the network and *v* denotes a one-hot encoding of the ground truth segmentation map. *u* and *v* have the shape *I* by *K* with *i* ∈ *I* being the voxels in the training data and *k* ∈ *K* being the classes. *u_i_,__**_k_* and *v_i_,__**_k_* represent the softmax probability output and ground truth for class *k* at voxel *i**,* respectively [40].

In addition to the weighted cross-entropy loss and multiclass dice loss function, Lin et al. introduced the focal loss method to add a modulating factor (1-*p_i_*)^γ^ to the cross-entropy loss with tunable focusing parameter γ ≥ 0 [47]. Focal loss is the improved version of the cross-entropy function. It assigns more weights to misclassified examples (e.g., background image) by increasing the importance of correcting. It is defined as Equation (3). Focal loss is used for semantic segmentation and medical object detection [48].
(3)$$FL(p_{t})=-{\left(1-p_{t}\right)}^{\lambda }\log(p_{t})$$

In a study, the embedding cost-sensitive naive Bayes (NB) classifier was used as a meta-learner to handle class imbalance for cancer gene expression data [49]. In another study, Shon et al. proposed a cost-sensitive hybrid deep learning approach for kidney cancer classification [50].

#### 3.2.3. Ensemble Learning-Based Methods

Ensemble learning combines multiple weak learners to acquire a robust model with better performance by using various mechanisms such as voting and boosting [51,52]. A decision is typically generated in supervised pattern recognition and machine learning tasks [52]. Ensemble learning models have a better performance than individual learning models, and they provide more resistance to noise/outlier. However, the training time for these learning models is time-consuming, and they can cause over-fitting in some cases [10].

Tang et al. constructed multi-omics data with the transcriptome and functional proteomics data and employed a bagging-based ensemble learning method on breast cancer to avoid over-fitting problems [53]. Le et al. utilized an ensemble of pre-trained ResNet50 CNN models for the classification of skin cancer to support dermatologists in skin cancer diagnosis [54]. Their model is based on a class-weighted learning approach for different classes in the loss function. Meanwhile, Wang et al. introduced a multi-layer perceptron (MLP)-based ensemble learning architecture for the classification of advanced gastric cancer (AGC) [55]. To reduce the class imbalance and keep the network compatible, they optimized the MLP with cross-entropy loss.

### 3.3. Hybrid Methods

Hybrid methods combine data-level methods and algorithm-level methods [15]. Data-level methods are utilized to process data externally and set the distribution of classes for instances. Afterward, the learning process is carried out internally with algorithm-level methods. With hybrid methods, the learning model will not be biased too much towards the majority class during the classification stage [56]. Detailed descriptions of these methods can be found in [15].

Random Under-Sampling Boost (RUSBoost) is one example of a hybrid method. To increase the predictive performance of colorectal cancer with the imbalanced data, Zhao et al. employed the RUSBoost algorithm for the colorectal cancer diagnosis [57]. A RUSBoost classifier was also used by Urdal et al. to solve and alleviate the class imbalance problems for the prognosis prediction of superficial urothelial carcinomas [58].

### 3.4. Other Methods

For omics and proteomic data in bioinformatics, machine learning-based models are generally faced with the rarity in the target class or class imbalance problem [59]. For example, a predictive model trained to predict the location of enhancers (i.e., positive samples) in the genome suffers from the class imbalance problem due to a large proportion of negative samples (i.e., non-enhancers) [59]. The prediction of contact-map or post-translation modifications (PTM) sites in a protein sequence, DNA methylation status/site prediction, identification of antimicrobial peptides (AMP) functional types, and the mutation incidence are one of the examples of class imbalance problems [59].

Different feature selection approaches and/or ensemble learning methods are employed to resolve this data imbalance problem. Feature selection approaches are considered dimensionality reduction techniques. Due to the high dimensionality of proteomic data, analyzing, storing, training, and classifying these data may be considered an NP-Hard problem [53,60]. Feature selection methods to find optimal feature subsets reduce the computational and storing cost and eliminate redundant and irrelevant information, facilitating data visualization. Several studies in oncology have adopted feature selection for cancer detection or diagnosis. Feature selection methods generally can be grouped into three categories: Filter, wrapper, and embedded methods (see Figure 4). Filter methods are independent of the chosen learning model, selecting variables by utilizing various statistical measures such as correlation. Wrapper methods assess the features according to the performance of the estimated model and selection criterion [61]. Embedded feature selection methods such as least absolute shrinkage and selection operator (LASSO) regression [62] perform the feature selection process as an integral part of the learning process, often specific to a provided predictor. Lao et al. proposed a deep learning-based radiomics model for survival prediction of Glioblastoma Multiforme (GBM) [66], wherein deep and handcrafted features were extracted from multi-modality MR images employing the LASSO regression model for the feature selection. Their deep imaging feature-based biomarker provides a potential for the preoperative care of GBM patients [66]. Tang et al. employed multi-omics data (transcriptome profiles and functional proteomics data) for breast cancer diagnosis [53], introducing a novel framework with hybrid feature selection and bagging-based ensemble learning methods to resolve the class imbalance problem. Wu et al. demonstrated that dosiomics improves predicting locoregional recurrence for head and neck cancer cases and should be proposed correlatively. They constructed the RadModel and RadDosModel for this purpose [67], extracted shape-based, statistical, texture-based radiomic features using Pyradiomics tool, and subsequently selected relevant and essential features which were then compressed using principal component analysis technique (PCA). Ensemble learning technique (EasyEnsemble) was applied to handle class imbalance problems.

Apart from these feature selection and ensemble learning methods, various class imbalance-specific evaluation measures are also being proposed. Conventional evaluation measures such as accuracy may not yield effective results in solving class imbalance problems as models may prove insufficiently trustworthy. For example, the accuracy rate may result in high classification performance in class imbalance problems in the event of the presence of a few rare samples, but it may not correctly classify these correctly. Thus, it may be necessary to assign more weight to false positive samples or false negative samples to improve the performance of the predictor instead of relying only on the accuracy rate. Class imbalance-specific evaluation measures such as the area under the precision-recall curve (auPRC), Matthews correlation coefficient (MCC), F-score, and the geometric mean (Gmean) of sensitivity and specificity [59,63] however may not yield adequate results. The equations for accuracy (*ACC*), precision (*PRE*), recall (*REC*), and F-score measures [64] are illustrated in Equation (4).
(4)$$ACC=\frac{TP+TN}{TP+TN+FP+FN}\;\;\;\;\;\;PRE=\frac{TP}{TP+FP}$$
$$REC=\frac{TP}{TP+FN}\;\;\;\;\;\;F-measure=\frac{2}{1/PRE+1/REC}$$
where *TP*, *TN*, *FP*, and *FN* represent the true positives, true negatives, false positives, and false negatives, respectively.

## 4. Guidelines to Mitigate Bias and Address Class Imbalance in Oncologic Data Sets towards Transferrable AI

### 4.1. Clinical Data

(1)Curate data with a view towards aggregation with parallel data sources that include the possibility to cross-check data collection against other data parameters, e.g., state or government data sources that can act as data verifiers of SDOH, disability, geographical location, and access to care.(2)Collect binary data when collecting more detailed data is not possible or impractical to obtain.(3)Collect data electronically with limited manual input and, when possible, automatically limit dependence on clinical provider input to allow for consistent long-term data collection as alternatives augment bias when some centers/settings collect data and others less so or not at all.(4)Implement data curation/clean data and data “health checks” for known sources of bias and class imbalance.(5)Implement capture and analysis of SDOH, disability, sex differences, ethnicity, and race data whenever possible and subject analysis to bias check, which can then be accounted for in future research and interpretation of data and resulting algorithms.

### 4.2. Other Data

Data-level, algorithm-level, and hybrid-level methods can mitigate bias and address class imbalance in oncologic data sets for imaging and omic, proteomic, and genomic data types. Researchers have at their disposal various feature selection/reduction techniques or ensemble learning-based methods that can be utilized for various types of bioinformatics data. In addition to this, diverse performance metrics such as the area under the precision–recall curve (auPRC), Matthews correlation coefficient (MCC), F-score, and the geometric mean (Gmean) of sensitivity and specificity can yield more effective results for class imbalance problems and should be considered when class imbalance is apparent. The most optimal method to be employed for a specific problem will ultimately depend on the data set characteristics (i.e., imbalance ratio) and the problem domain (e.g., the degree of importance, the main objective). Several approaches need to be examined to arrive at the best approach, particularly when a novel hypothesis or novel data set is involved, and the data supporting the most optimal approach may be lacking. We advise experimenting with different combinations, including traditional approaches, before arriving at the most optimal scheme in order to obtain more robust, efficient transferrable results for the related model and data set. However, if the results do not improve, data augmentation/reduction techniques can also be applied.

## 5. Conclusions

Acknowledging the complexity of the technology required to advance AI in medicine, clinicians represent a set of stakeholders most likely to appreciate potential sources of information bias in data sets. However, technical resources are most likely to possess the capability to mitigate algorithmic bias, while both disciplines are invariably involved in recognizing and addressing pervasive knowledge translation bias. This review examined and summarized diverse methods to handle class imbalance problems for AI in medical settings, particularly oncologic data. Some methods rely on accounting for data distribution, while others relate to modifying the learning model. Because there is no universally accepted solution for all problems, every approach has advantages and disadvantages related to the data set, research question, and learning model used. As a future direction of this work, we aim to improve computer-aided diagnosis (CADx) results and the management of malignancy by employing different and efficient approaches for handling the class imbalance problems in AI. CADx provides unique opportunities to identify biological triggers for tumor response and resistance by leveraging large scale data that may connect to existing scientific knowledge in fascinating and practice changing ways while also posing great challenges to understanding a wealth of growing data while filtering meaningful signals from the noise while conserving and growing crucial pattern recognition to truly positively alter outcomes.

## Figures and Tables

**Figure 1 cancers-14-02897-f001:**
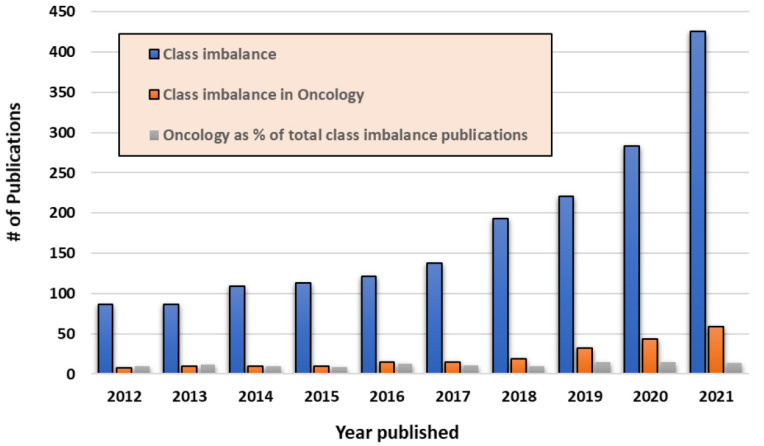
The number of publications on class imbalance listed in Pubmed by year for years 2012 to 2021 (blue = total publications per year on class imbalance, orange = total publications per year on class imbalance in oncology, grey = the number of publications on class imbalance pertaining to oncology as a percent of the total number of publications that year) [8].

**Figure 2 cancers-14-02897-f002:**
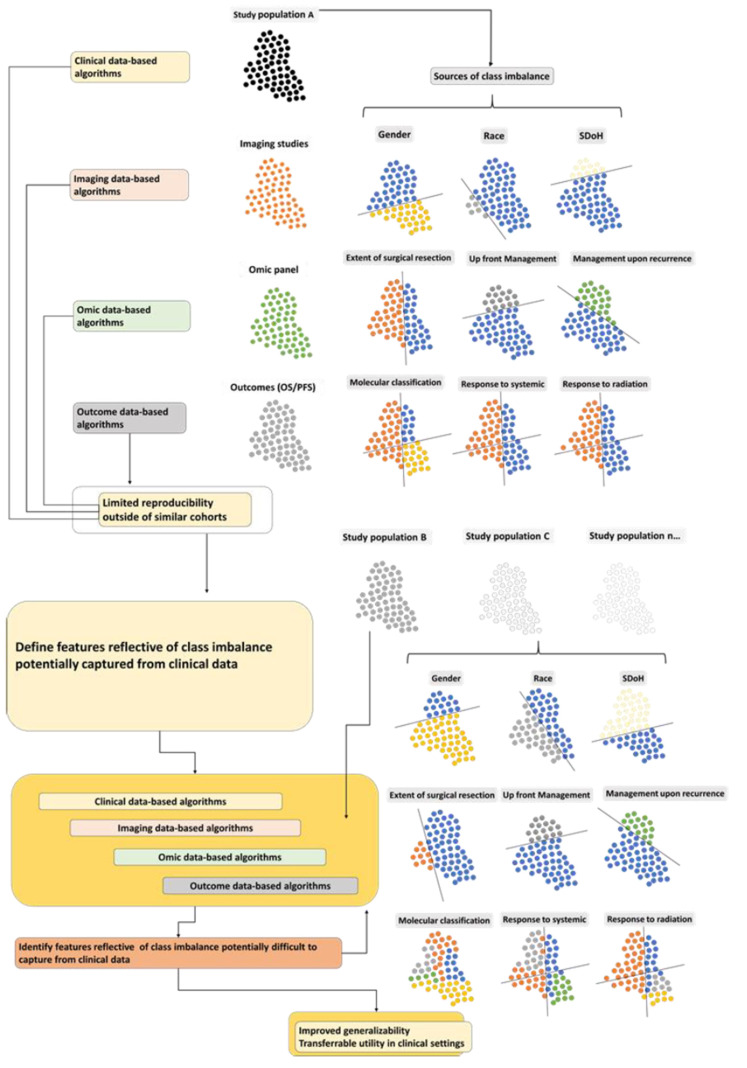
Clinical, imaging, omic and outcome based algorithms (**left panel**) are generated based on data sets that harbor underrepresentation or overrepresentation of one or several categories as related to demographics (gender, race, social determinants of health–(**right top panel**), management (extent of surgical resection, type of upfront management, management upon recurrence (**right middle panel**) and disease characteristics (molecular classification, response to systemic management and radiation therapy (**right lower panel**). Class imbalance affects ancillary data sets such as imaging, omics based on biospecimens and outcomes (**middle panel**). The resulting algorithms result in limited reproducibility in other cohorts (Study population, B, C, and a theoretical infinite other population that does not share identical class imbalance). As both the sources and extent of over and underrepresentation of certain classes and class imbalance are altered in other populations/data sets (**lower panel**), defining features reflective of class imbalance and mitigating these via compensatory methods in all data subtypes, can address the lack of reproducibility and help identify additional features that can be used to further optimize algorithms allowing for transferrable results (**lower panel**).

**Figure 3 cancers-14-02897-f003:**
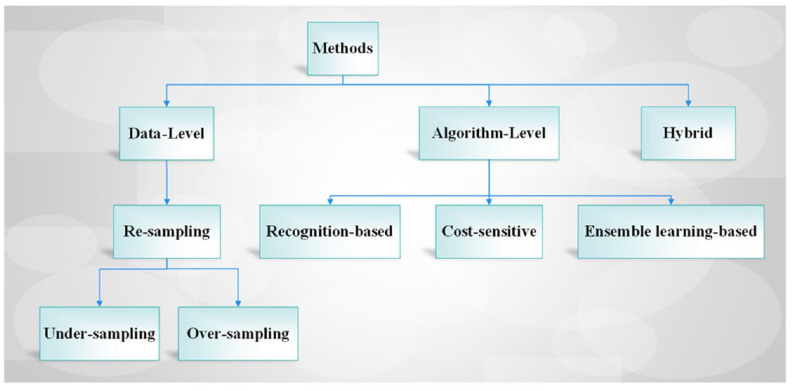
An overview of the existing methods for the class imbalance problem.

**Figure 4 cancers-14-02897-f004:**
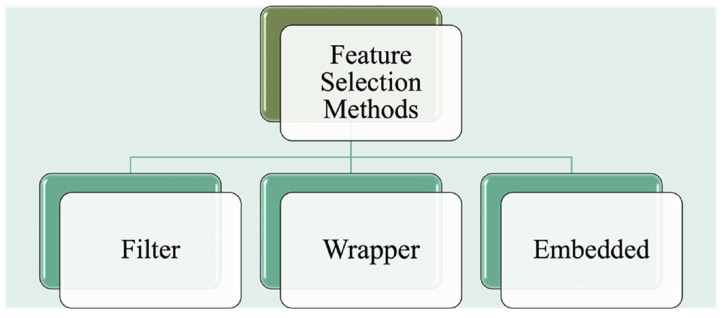
The overview of the feature selection methods.

**Table 1 cancers-14-02897-t001:** Pertinent literature addressing approaches to class imbalance.

Study	Technique	Setting
**Data-level methods**		
Bose et al. [35]	Convolutional neural networks (CNN), long short-term memory (LSTM), over-sampling	Radiation Oncology, incident reporting
Brown et al. [36]	Under-sampling	Radiation Oncology, prostate cancer
Liu et al. [37]	Synthetic Minority Oversampling Technique (SMOTE)	Glioblastoma imaging
Suarez-Garcia et al. [38]	Under-sampling	Glioma imaging
Li et al. [39]	Under-sampling, K-Means++ and learning vector quantization (LVQ)	Liver cancer
Isensee et al. [40]	Deep learning, nnU-net	Brain tumor segmentation
**Algorithm level methods**		
Goyal et al. [41]	Recognition-based, one class classification	real-world data sets across different domains: tabular data, images (CIFAR and ImageNet), audio, and time-series
Gao et al. [42]	Recognition-based, one class classification	Medical imaging
Welch et al. [43]	Recognition-based	Head and Neck Radiation therapy
Leevy et al. [44]	Cost-sensitive	General
Nguyen et al. [45]	Cost-sensitive, comparison of techniques	SMOTE and Deep Belief Network (DBN) against the two cost-sensitive learning methods (focal loss and weighted loss) in churn prediction problem
Milletari et al. [46]	Cost-sensitive	Medical image segmentation
Lin et al. [47]	Cost-sensitive, Focal loss	Medical imaging
Jaeger et al. [48]	Cost-sensitive, Focal loss	Medical imaging object detection
Xiong et al. [49]	Cost-Sensitive Naive Bayes Stacking Ensemble	Various malignancy data sets and data types
Shon et al. [50]	Cost-sensitive	Kidney cancer data (TCGA)
Dong et al. [51]	Ensemble-Learning	General
Sagi et al. [52]	Ensemble-Learning	General
Tang et al. [53]	Ensemble-Learning, bagging-based	Transcriptome and functional proteomics data breast cancer
Le et al. [54]	Ensemble-Learning, ResNet50 CNN	Skin cancer
Wang et al. [55]	Ensemble-Learning, multi-layer perceptron (MLP)-based	Gastric cancer
**Hybrid methods**		
Khushi et al. [15]	Comparative Performance Analysis of Data Re-sampling Methods	Various malignancy data sets and data types
Chen et al. [56]	Combination of methods	General
Zhao et al. [57]	Random Under-Sampling Boost (RUSBoost)	Colorectal cancer, microarray data
Urdal et al. [58]	Random Under-Sampling Boost (RUSBoost)	Urothelial carcinoma, histopathology data
**Other methods and reviews**		
Mirza et al. [59]	Integrative analysis of biomedical big data	
Guyon et al. [60]	variable and feature selection overview	
Hilario et al. [61]	Class imbalance in proteomic biomarker studies overview	
Tibshirani et al. [62]	LASSO (Least Absolute Shrinkage and Selection Operator) overview	
Yan et al. [63]	Graph- and kernel-based—omics data integration algorithms	
Fawcett et al. [64]	ROC analysis overview	

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
