# Peer review of "Bias and Class Imbalance in Oncologic Data—Towards Inclusive and Transferrable AI in Large Scale Oncology Data Sets"

_cancers, 2022, doi:10.3390/cancers14122897_

Round 1

Reviewer 1 Report

Erdal et al provide a nice review on the bias and imbalance data issues for machine learning tasks in oncology and the content of the review is adequate in summarizing different aspects of data imbalance problem as well as strategies to handle with it. An in-depth understanding of data bias and imbalance and proper handling of these issues can help foster awareness of researchers and clinicians to better design their clinical experiments as well as to avoid bias or wrong interpretation due to potential confounding factors missing in the clinical data. In my opinion, the manuscript is well written. Only minor concerns as below:

1.     Since the audience of the journal is mainly oncologists, it is necessary to provide a simple but clear definition for class imbalance at the beginning of the Introduction and explain why it matters from clinical perspectives.

2.     Although the authors define class imbalance as “The degree of imbalance is defined as the ratio of the sample size of the minority class to that of the majority class” in page 2, it is still not clear what does the authors mean by “minority” and “majority” classes. Elaborate with some examples will be helpful.

3.     Please provide further elaboration on challenges and opportunities for improving computer-aided diagnosis (CADx) at Conclusion section.

4.     The resolution of figures seems poor, please improve the resolution.

Author Response

Response to Reviewer 1

  1. Since the audience of the journal is mainly oncologists, it is necessary to provide a simple but clear definition for class imbalance at the beginning of the Introduction and explain why it matters from clinical perspectives.

We agree with this point by the reviewer and have added a definition for class imbalance in the introduction to address this.

  1. Although the authors define class imbalance as “The degree of imbalance is defined as the ratio of the sample size of the minority class to that of the majority class” in page 2, it is still not clear what does the authors mean by “minority” and “majority” classes. Elaborate with some examples will be helpful.

We agree with this point by the reviewer and in the introduction, we have added a definition that better connects class imbalance to unequal distributions within data sets with associated examples in oncologic data sets and defined majority and minority classes.

  1. Please provide further elaboration on challenges and opportunities for improving computer-aided diagnosis (CADx) at Conclusion section.

We have corrected this by elaborating on this concept in the conclusion.

  1. The resolution of figures seems poor, please improve the resolution.

A higher resolution file has been submitted for each figure as well as additional versions for figure 2 to allow for most optimal display in the final publication.

Reviewer 2 Report

The review article aims to provide the details of the class imbalance problems in cancer datasets and are also applicable to any other medical diagnostic data, especially when the data is high dimensional and is almost impossible to avoid such bias. Moreover, review article allows the authors in the AI based analysts to carefully consider all possible bias, the general recommendations and examples that were applied in the earlier studies. The manuscript is written very well. However, I have few minor comments which can improve the overall quality of the manuscript.

comments

  1. To me Figure 2 is not clear and rather not explained very well. To me it sounds figure 2 is the major figure in the review article describing the overall problems. To those who are not expertise in the field, the figure is difficult to understand. Can the authors add more description in the figure legend and make it more understandable?
  2. Regarding resampling, there can be cases where the resampling may not provide performance of the prediction models. Can the authors bring these details into the review? Possibly any explanation and recommendations for the same?
  3. The article has been written in good English, however, it will be good to have a English language check for once, to avoid any minor spelling or sentence mistakes.

Author Response

Response to Reviewer 2

  1. To me Figure 2 is not clear and rather not explained very well. To me it sounds figure 2 is the major figure in the review article describing the overall problems. To those who are not expertise in the field, the figure is difficult to understand. Can the authors add more description in the figure legend and make it more understandable?

We agree with the reviewer and have added further explanation to improve clarity. A higher resolution file has been submitted as well as additional versions to allow for most optimal display in the final publication.

  1. Regarding resampling, there can be cases where the resampling may not provide performance of the prediction models. Can the authors bring these details into the review? Possibly any explanation and recommendations for the same?

We have added a paragraph addressing this point in section 3.1.

  1. The article has been written in good English, however, it will be good to have a English language check for once, to avoid any minor spelling or sentence mistakes.

A Grammarly check has been carried out and corrections applied prior to submission and repeated with version.

Reviewer 3 Report

The authors critically review methods to deal with bias and class imbalance in machine learning-based problems, with a focus on oncology.

The issue of class imbalance is clearly illustrated and dealt with. I think the concept of bias is a little confusing because sometimes authors refer to the machine learning taxonomy for bias (i.e., the difference between actual and predicted values), and others to bias in the epidemiological sense. For example, they cite gender bias, which refers to an inadequate representation of men and women in study samples and, in a broader sense, to neglected differences in biological mechanisms between males and females.

The definition and discussion about bias should be better circumstantiated.

Author Response

Response to Reviewer 3

  1. The issue of class imbalance is clearly illustrated and dealt with. I think the concept of bias is a little confusing because sometimes authors refer to the machine learning taxonomy for bias (i.e., the difference between actual and predicted values), and others to bias in the epidemiological sense. For example, they cite gender bias, which refers to an inadequate representation of men and women in study samples and, in a broader sense, to neglected differences in biological mechanisms between males and females.

 We agree and have added additional content in the introduction to clarify this aspect. We have made the distinction and connection between data and algorithm related bias more clear by defining both terms and connecting the concepts more clearly in the introduction and in section 2 addressing the point raised by the reviewer.

  1. The definition and discussion about bias should be better circumstantiated.

We agree with this point by the reviewer and have corrected this in the introduction and in section 1.